# Protective Effect of Melatonin Against Bisphenol A Toxicity

**DOI:** 10.3390/ijms26157526

**Published:** 2025-08-04

**Authors:** Seong Soo Joo, Yeong-Min Yoo

**Affiliations:** 1Department of Marine Bioscience, College of Life Science, Gangneung-Wonju National University, Gangneung 25457, Republic of Korea; ssj66@gwnu.ac.kr; 2Institute of Environmental Research, Kangwon National University, Chuncheon 24341, Republic of Korea

**Keywords:** BPA toxicity, melatonin, protective effect, mechanism of action, potential preclinical applications

## Abstract

Bisphenol A (BPA), a prevalent endocrine-disrupting chemical, is widely found in various consumer products and poses significant health risks, particularly through hormone receptor interactions, oxidative stress, and mitochondrial dysfunction. BPA exposure is associated with reproductive, metabolic, and neurodevelopmental disorders. Melatonin, a neurohormone with strong antioxidant and anti-inflammatory properties, has emerged as a potential therapeutic agent to counteract the toxic effects of BPA. This review consolidates recent findings from in vitro and animal/preclinical studies, highlighting melatonin’s protective mechanisms against BPA-induced toxicity. These include its capacity to reduce oxidative stress, restore mitochondrial function, modulate inflammatory responses, and protect against DNA damage. In animal models, melatonin also mitigates reproductive toxicity, enhances fertility parameters, and reduces histopathological damage. Melatonin’s ability to regulate endoplasmic reticulum (ER) stress and cell death pathways underscores its multifaceted protective role. Despite promising preclinical results, human clinical trials are needed to validate these findings and establish optimal dosages, treatment durations, and safety profiles. This review discusses the wide range of potential uses of melatonin for treating BPA toxicity and suggests directions for future research.

## 1. Introduction

### 1.1. Overview of BPA Toxicity

Bisphenol A (BPA) has been widely used in the manufacturing of polycarbonate plastics and epoxy resins since its synthesis in the 1950s [1]. BPA is incorporated into various everyday products, such as food and beverage containers, water bottles, epoxy linings for canned goods, bottle caps, water pipes, shatterproof windows, eyewear, dental sealants, and medical devices [1]. Humans are widely exposed to BPA, which primarily occurs through the diet owing to the ability of BPA to migrate from food and drink containers, especially under heat [1,2]. Measurable levels of BPA have been detected in the vast majority of the population, raising public health concerns, especially in vulnerable groups such as fetuses and infants [1]. BPA is classified as an endocrine-disrupting chemical due to its profound capacity to interfere with the body’s delicate hormonal signaling pathways [2,3]. This disruption occurs because BPA can mimic or antagonize the actions of natural hormones by binding to critical hormone receptors. Specifically, it interacts with estrogen receptors (ERα/ERβ), where it can either enhance or diminish the physiological effects of estrogen, a hormone vital for development, reproduction, and metabolic regulation. Furthermore, BPA binds to androgen receptors, thereby interfering with male reproductive processes and other androgen-dependent functions. Its disruptive influence also extends to thyroid hormone receptors, potentially unsettling the intricate balance of thyroid hormones crucial for metabolism, growth, and neurodevelopment [2,3]. Consequently, these broad and pervasive disruptions to hormonal regulation are closely associated with a wide array of adverse health effects, encompassing concerns such as reproductive issues, metabolic disorders, neurodevelopmental impacts, and an increased susceptibility to hormone-sensitive cancers [4]. The European Food Safety Authority (EFSA) has substantially lowered the tolerable daily intake of BPA in response to the growing scientific evidence of the effects of BPA and heightened concerns regarding its health effects [5,6]. Consequently, regulatory agencies are intensifying their efforts to limit BPA exposure and promote the development of safer alternatives for consumer products.

### 1.2. Overview of Melatonin and Its Potential Protective Effects

Melatonin is a neuroendocrine hormone that is mainly produced and released by the pineal gland and plays a central role in controlling circadian rhythms and the sleep–wake cycle [7]. Melatonin exerts strong antioxidant and anti-inflammatory effects in addition to its chronobiological functions [7]. Melatonin functions as a direct scavenger of reactive oxygen and nitrogen species (ROS and RNS, respectively); melatonin boosts the activity of key endogenous antioxidant enzymes, such as superoxide dismutase (SOD), catalase (CAT), and glutathione peroxidase (GPx) [8]. The anti-inflammatory actions of melatonin are mediated via the downregulation of pro-inflammatory cytokines and the inhibition of key inflammatory signaling pathways [7]. Melatonin has been studied as an agent protecting against the toxicity induced by various environmental pollutants, such as BPA, because of these properties [9]. Melatonin mitigates the genotoxic effects of BPA by reducing the expression of oxidative stress markers and strengthening cellular antioxidant defenses, which indicates the potential of melatonin as a therapeutic intervention in BPA-related toxicity [9]. These findings suggest that melatonin plays a crucial protective role in vulnerable populations exposed to BPA, helping to preserve cellular integrity and function through attenuating oxidative stress and inflammation.

Melatonin is also produced and metabolized in peripheral tissues [10] and is found in a number of nutritional products, including honey [11]. It is important to note that many of the effects of melatonin may be secondary to the actions of its metabolites [12], the metabolism of melatonin is very rapid [13], and the half-life of melatonin in serum is short [10].

### 1.3. Purpose 

Our aim in this review was to summarize the current evidence on the protective effects of melatonin against BPA toxicity, drawing from recent animal, in vitro, and limited clinical studies. Firstly, a comprehensive review of the mechanisms underlying BPA-induced toxicity was conducted. The present study examined the potential of using melatonin to counteract the effects of BPA through its antioxidant, anti-inflammatory, and cytoprotective properties. Finally, the potential clinical applications of melatonin were discussed, with the indication of key directions for future research.

## 2. BPA Toxicity

### 2.1. Sources of BPA Exposure

Human exposure to BPA is widespread and primarily occurs by ingesting contaminated food and beverages [1]. BPA can migrate from epoxy-coated metal cans and polycarbonate plastic containers, especially when subjected to heat, such as during microwaving or dishwashing [1,2]. Nondietary exposure occurs through thermal paper (e.g., cash register receipts), dust, air, water, medical devices, and dental sealants. Occupational exposure to BPA is common among workers involved in manufacturing BPA-containing products or the handling of thermal paper [1,2]. BPA has been detected in human breast milk, indicating that lactation may constitute a potential route of BPA exposure for infants [1]. Furthermore, BPA exposure has been associated with various adverse health outcomes in adult men and women, including pregnant women, highlighting its broad relevance to public health across different life stages [14,15,16]. Complete avoidance of BPA remains a considerable challenge, given the pervasive presence of BPA in everyday products.

### 2.2. Mechanisms of BPA Toxicity

BPA primarily exerts toxic effects by acting as an endocrine disruptor and interacting with multiple hormone receptors [2,3]. These interactions include the (1) binding to estrogen receptors (ERα/ERβ), which, at concentrations as low as 1 nmol/L alter the balance between cell proliferation and apoptosis [17,18]; (2) activating of androgen receptors, which play a crucial role in male sexual differentiation during fetal development [18,19]; (3) disrupting thyroid hormone receptors, particularly by interfering with T3-mediated signaling through integrin αvβ3/MAPK pathways [17]; and (4) influencing G-protein-coupled estrogen receptors, which impact calcium homeostasis in pancreatic cells [17,18]. These BPA–receptor interactions contribute to the development of a range of health issues such as metabolic disorders, reproductive abnormalities, and increased cancer risk [18]. These disruptions modulate gene expression and cellular functions, contributing to the overall toxicity of BPA [2,3].

BPA triggers oxidative stress by increasing the production of ROS as well as RNS and weakening antioxidant defenses [2,3,20]. Oxidative stress reduces the mitochondrial membrane potential in human cell models, particularly intestinal Caco-2 cells [21]. Other BPA derivatives, such as bisphenols AF, AP, and P, exhibit equal to or higher toxicity than BPA in human cell models, with dose-dependent increases in ROS (150% above control levels in Caco-2 cells) [16] and the disruption of mitochondrial calcium homeostasis [21].

BPA compromises mitochondrial function and triggers epigenetic alterations, such as modifying DNA methylation and microRNA expression [2,3]. Additionally, BPA initiates various forms of cell death, including apoptosis, necroptosis, pyroptosis, ferroptosis, and autophagy, in different cell types [4], which may contribute to the disruption of gut microbiota [2,3]. In particular, the immune system is highly sensitive to BPA exposure, which strongly affects immune function [2,3].

BPA has genetic and epigenetic effects. BPA exposure damages DNA through the p53–p21 pathway [20] and reduces global DNA methylation by suppressing DNMT gene expression [19,22]. BPA also hypermethylates specific genes, such as those linked to prostate cancer, by altering histone [17,22].

Figure 1 summarizes the mechanisms of BPA toxicity described above.

### 2.3. Effects of BPA Exposure on Health

BPA exposure has been strongly associated with a wide range of harmful health effects in animal and human studies [14,15,16]. BPA mimics estrogen by binding to estrogen receptors, a process that is linked to adverse cardiovascular effects such as impaired cardiac excitability, oxidative stress, and epigenetic modifications. BPA exposure increases the risk of cardiovascular diseases, including hypertension, atherosclerosis, and diabetes, particularly during sensitive periods, such as pregnancy [23,24,25]. However, the epidemiological evidence of the effects of BPA is inconsistent, with some attributing these inconsistencies to limitations in exposure assessments and study designs [26].

BPA is an obesogen that contributes to increased body weight, adipogenesis, and lipid dysregulation. The results of epidemiological and experimental studies suggest a correlation between BPA exposure and obesity, as well as metabolic disorders, particularly during critical developmental windows such as the prenatal and early childhood stages [27,28]. However, the precise mechanisms underlying the effects of BPA remain unclear, with some studies yielding inconclusive or conflicting results [26].

The evidence is increasingly suggesting that BPA exposure is linked to reproductive health concerns, including hormonal disruption and potential fertility issues. Prenatal BPA exposure is associated with altered reproductive hormone levels in neonates, having sex-specific effects [29,30]. Additionally, BPA exposure during development disrupts sexually selected traits and cognitive abilities in animal models, raising concerns about its impact on human development [31].

Recent evaluations by regulatory bodies such as the EFSA have highlighted that the immune system is particularly sensitive to BPA exposure. The effect of BPA on Th17 cells, which play a role in inflammatory and autoimmune conditions, has been emphasized. The EFSA identified immune toxicity as the most sensitive endpoint in 2023, lowering the tolerable daily intake to 0.2 ng/kg/day, a reduction of 20,000 times compared with that in previous guidelines [5]. The key mechanisms through which the immune system is disrupted include T-helper 17 cell activation promoting autoimmune responses [22], increased susceptibility to lung inflammation through altered immune cell populations [32], and mitochondria-mediated apoptosis in leukocytes at environmentally relevant doses (1–100 μg/mL) [22].

BPA binds to estrogen receptors (ERα/ERβ), thyroid hormone, and G-protein-coupled estrogen receptors at concentrations as low as 1 nmol/L, altering cellular proliferation, calcium signaling, and metabolic pathways [22]. These BPA–receptor interactions contribute to reducing fertility and ovarian dysfunction in women, with higher serum BPA levels associated with a higher prevalence of polycystic ovary syndrome [22,33]. BPA exposure has been linked to testicular atrophy and reduced sperm quality in men [22], whereas developmental abnormalities have been identified in fetal mammary gland tissue after BPA exposure [32,33].

Prenatal BPA exposure has been associated with the early onset of puberty and behavioral changes during adolescence [33]. Perinatal exposure to BPA doses of ≤1 μg/kg/day disrupt the development of the hypothalamic–pituitary–gonadal axis [33]. Additionally, a 72 h embryonic exposure of mice to BPA reduces blastocyst formation rates by 40%, likely due to oxidative DNA damage [22].

Human exposure to BPA is widespread, primarily through dietary sources, such as food packaging and canned goods. Biomonitoring studies have consistently detected BPA in the urine, blood, and other tissues, indicating pervasive BPA exposure. However, the discrepancies in the findings between biomonitoring and toxicokinetic studies have raised questions regarding the actual internal BPA exposure levels and their health implications [34,35].

The health effects of BPA exposure are complex and multifaceted, including endocrine disruption, cardiovascular risks, metabolic disorders, reproductive dysfunction, and immune system sensitivity. Although our understanding of these effects has substantially progressed, the inconsistencies in study designs and exposure assessments underscore the need for further research to clarify the long-term risks and mechanisms of BPA toxicity.

Table 1 summarizes the findings of recent studies on the association between BPA exposure and various types of cancer. The data were extracted from studies published between 2020 and 2025, emphasizing the most recent publications.

## 3. Melatonin and Its Protective Effects

### 3.1. What Is Melatonin?

Melatonin (N-acetyl-5-methoxytryptamine) is an evolutionarily conserved neurohormone predominantly produced in the pineal gland from the amino acid tryptophan [51,52]. Melatonin production follows a circadian rhythm, with low and high levels during the day and night, respectively, regulating the sleep–wake cycle of the body [53,54]. 

Melatonin exerts its effects through binding to G-protein-coupled membrane-bound receptors, MT1 and MT2, which are encoded by Mtnr1a and Mtnr1b, respectively. These receptors are widely distributed in various tissues, such as the hypothalamus and retina. Activating the MT1 and MT2 receptors promotes sleep and regulates circadian rhythms [54,55]. 

Melatonin is produced in various tissues and organs in addition to the pineal gland, including the retina, gastrointestinal tract, platelets, skin, bone marrow, lymphocytes, ovaries, and testes. Melatonin may exert local (autocrine or paracrine) effects in these areas, which influence various physiological processes [56,57].

### 3.2. Antioxidant Properties of Melatonin

Melatonin is a potent antioxidant that acts through direct and indirect mechanisms [7]. Melatonin directly scavenges a broad range of free radicals, including hydroxyl radicals, superoxide anions, hydrogen peroxide, peroxynitrite, and singlet oxygen, effectively terminating radical chain reactions [8,55]. The metabolites of melatonin also exhibit antioxidant activity and contribute to the formation of an antioxidant cascade [58]. Melatonin enhances the activity and expression of endogenous antioxidant enzymes such as SOD, CAT, and GPx. This effect appears to indirectly occur through mechanisms that have not yet been fully elucidated [7,8,9,55,58,59]. Furthermore, melatonin inhibits pro-oxidative enzymes such as nitric oxide synthase (NOS) and lipoxygenase, chelates metal ions that catalyze oxidation, and protects mitochondrial function through reducing oxidative stress and preventing mitochondrial DNA damage [7,60].

### 3.3. Anti-Inflammatory Properties of Melatonin

Melatonin exhibits anti-inflammatory effects through modulating various components of the inflammatory response [7,61,62]. Melatonin inhibits the production and release of pro-inflammatory cytokines such as interleukin (IL)-1β, IL-6, and IL-8, as well as tumor necrosis factor-α (TNF-α) [60,61,62]. Additionally, melatonin interferes with the nuclear translocation of nuclear factor-kappa B (NF-κB), a key transcription factor involved in upregulating inflammatory cytokines [60]. Melatonin downregulates the expression of pro-inflammatory enzymes such as cyclooxygenase-2 and inducible nitric oxide synthase and reduces the production of adhesion molecules that facilitate leukocyte migration [60,63]. Moreover, melatonin promotes macrophage polarization toward the anti-inflammatory M2 phenotype [63]. Although the role of melatonin in inflammation may be context-dependent and vary with the stage of the inflammatory process, melatonin generally exerts anti-inflammatory effects under chronic inflammatory conditions [59,60].

Several experimental and clinical studies have described the anti-inflammatory effects of melatonin. Melatonin has been found to reduce the inflammation associated with arthritis, colitis, and sepsis in animal models [64,65]. Similarly, melatonin has been found to reduce the levels of inflammatory markers in patients with conditions such as rheumatoid arthritis and periodontitis [66,67]. These anti-inflammatory properties are hypothesized to contribute to the protective effects of melatonin against BPA toxicity because inflammation is implicated in the pathogenesis of the adverse health effects induced by BPA.

### 3.4. Other Mechanisms Underlying Protective Effects of Melatonin

Melatonin is involved in several additional mechanisms that may contribute to its protective effects against BPA toxicity, in addition to its antioxidant and anti-inflammatory effects [68]. Melatonin directly interacts with various proteins, including enzymes involved in detoxification and cellular signaling pathways [69]. Melatonin influences epigenetic regulation by modulating the expression of miRNAs, affecting gene expression under various pathological conditions [69].

Melatonin plays a role in regulating mitochondrial dynamics, which are essential for maintaining mitochondrial function and cellular homeostasis [69]. Melatonin counteracts the Warburg effect in cancer cells by inhibiting aerobic glycolysis [69]. Melatonin interacts with calcium-binding proteins such as calmodulin and tubulin, influencing cytoskeletal organization and the related cellular processes [69]. In addition, melatonin binds to calreticulin, a protein involved in calcium homeostasis [69]. These diverse molecular interactions underscore the multifaceted protective role of melatonin at the cellular level.

Melatonin modulates several key signaling pathways, such as the cAMP/PKA, PI3K/Akt, MAPK/ERK, ERK/NF-κB [70], and p53/PUMA/Drp1 pathways [71], which may contribute to its protective effects under various cellular stress conditions. Collectively, these mechanisms suggest that melatonin confers broad protective effects against a range of toxic insults, such as BPA-induced toxicity.

Melatonin and its metabolites have been identified as agonists of the aryl hydrocarbon receptor and peroxisome proliferator-activated receptor gamma [72]. This suggests that melatonin may exert its effects through additional mechanisms beyond the classical MT1 and MT2 receptors. Through these alternative pathways, melatonin may confer cellular protection, including antioxidant and radioprotective effects, with particular potential for pharmacological applications in the prevention and treatment of skin diseases [72].

Table 2 and Figure 2 summarize the protective mechanisms of melatonin as reported in peer-reviewed scientific articles published between 2020 and 2025, with an emphasis on more recent studies. 

## 4. Evidence of Protective Effects of Melatonin Against BPA Toxicity

### 4.1. In Vitro Studies

Recent in vitro studies have demonstrated that melatonin exerts broad cytoprotective and genoprotective effects against BPA-induced toxicity across various cell types [9]. In human gingival fibroblasts, colon cancer cells, and bone marrow-derived stem cells, BPA exposure led to elevated oxidative stress and reduced cell viability. Co-treatment with melatonin not only restored cell viability, but also effectively prevented DNA damage in all three models [9]. Supporting these findings, comet assay results indicated that melatonin significantly reduced DNA strand breaks, as evidenced by a decreased tail moment, thereby mitigating BPA-induced genotoxicity [9].

At the mechanistic level, BPA-induced cytotoxicity and genotoxicity are primarily mediated through oxidative stress and endoplasmic reticulum (ER) stress. BPA increases ROS production, promotes lipid peroxidation, and depletes intracellular GSH, which are hallmarks of oxidative damage [9]. Melatonin counteracts these effects by scavenging ROS, restoring GSH levels, and reducing MDA concentrations across multiple in vitro systems [9].

Furthermore, BPA induces ER stress by upregulating molecular chaperones such as GRP78 and GRP94, and by activating unfolded protein response (UPR) signaling pathways via PERK, IRE1, and ATF6, ultimately leading to apoptosis [69]. Melatonin alleviates ER stress in various cellular models, including neuronal, hepatic, testicular, and osteoarthritic chondrocytes, by downregulating ER stress markers (e.g., phospho(p)-PERK, p-IRE1, XBP-1, GRP78, and GRP94) and by upregulating its own receptors, MT1 and MT2, particularly in testicular cells, thereby preserving cellular homeostasis and inhibiting apoptotic pathways [69].

Melatonin also plays a key role in maintaining mitochondrial integrity. Due to its high membrane permeability, it readily penetrates mitochondrial membranes, where it reduces oxidative burden and preserves mitochondrial bioenergetics in BPA-exposed cells [9].

A systematic review and meta-analysis of in vitro studies further substantiated melatonin’s superior protective efficacy compared with that of certain vitamins against BPA-induced reproductive toxicity. Molecular docking analyses demonstrated that melatonin has a stronger binding affinity to androgen and estrogen receptors than some vitamins, which themselves may exhibit endocrine-disrupting properties [84,85].

In summary, in vitro findings consistently demonstrate that melatonin ameliorates BPA-induced oxidative stress, DNA damage, ER dysfunction, and mitochondrial impairment through its potent antioxidant activities and modulation of ER stress and UPR pathways [9,69,85]. These observations support melatonin’s potential as a cytoprotective and genoprotective agent and warrant further in vivo investigations to validate its mechanisms and preclinical relevance.

### 4.2. Animal and Preclinical Studies

Numerous animal studies have investigated the protective effects of melatonin against BPA-induced reproductive toxicity [85]. In rodent models, melatonin administration alleviates BPA-induced testicular damage by promoting seminiferous tubule development, restoring germ cell organization, and increasing the thickness of the tubular epithelium [85]. Additionally, melatonin upregulates the expression of its receptors, MT1 and MT2, in BPA-exposed testicular tissue, thereby enhancing tissue sensitivity to its protective actions [84]. When co-administered with BPA, melatonin significantly improves key fertility parameters, such as sperm viability, sperm density, and serum testosterone levels, while concurrently reducing ER stress and apoptosis in testicular cells [69,85].

A 2023 systematic review and meta-analysis of animal studies confirmed that melatonin therapy attenuates BPA- and other pollutant-induced alterations in testicular histopathology, reproductive hormone levels, and oxidative stress markers [84,85]. Related investigations into melatonin’s interaction with BPA analogues on bone tissue have shown that treatment outcomes vary depending on dose, duration, and sex, underscoring the complexity of these interactions [84].

Central to melatonin’s protective mechanism is its ability to enhance endogenous antioxidant defenses. It significantly increases the activity of key antioxidant enzymes, such as SOD and GPx, while decreasing the levels of malondialdehyde (MDA), a lipid peroxidation marker, thereby reducing oxidative stress [86,87]. In FLKBLV cells exposed to BPA, melatonin reversed oxidative damage and apoptosis, in part through the activation of autophagy and inhibition of the p38 MAPK signaling pathway [88].

Beyond its antioxidant activity, melatonin exhibits anti-inflammatory effects by downregulating pro-inflammatory cytokines in BPA-exposed animal models, leading to reduced tissue damage and improved histological features [85]. This anti-inflammatory function is closely linked to melatonin’s mitochondrial protective effects. For example, in porcine oocytes, melatonin reduced mitochondrial superoxide production, promoted oocyte maturation, and prevented mitochondrial-mediated apoptosis following BPA exposure [89,90].

Preclinical studies in zebrafish further support melatonin’s role in mitigating BPA-induced developmental toxicity, including morphological abnormalities and disruptions in sleep–wake cycles during embryogenesis [91]. Collectively, these findings present melatonin as a multifaceted therapeutic candidate against BPA toxicity, operating through antioxidant defense, inflammation suppression, mitochondrial preservation, and reproductive function support.

Despite these promising preclinical results, well-controlled human trials remain limited. Indirect clinical evidence suggests that BPA exposure increases ROS and lipid peroxidation while depleting intracellular GSH levels in humans [9,92,93,94]. In preclinical studies, melatonin supplementation has been shown to counteract these biochemical disturbances by reducing ROS and MDA levels and restoring the activity of GSH, SOD, and CAT [9,88,89,90,91], supporting its translational potential in protecting vulnerable organs such as the liver, kidneys, and reproductive tissues.

Both animal and preclinical studies consistently highlight melatonin’s efficacy in preventing BPA-induced reproductive dysfunction. In males, melatonin protects testicular cells from apoptosis and improves sperm count and motility [9,69,84,93]. In females, melatonin reduces BPA-induced ovarian morphological abnormalities and helps to normalize the estrous cycle [9,84]. These effects are mediated through melatonin’s antioxidant properties, its ability to reduce DNA damage, and its regulation of ER stress pathways, including the downregulation of PERK, IRE1, and ATF6, and their downstream effectors [69].

Nevertheless, substantial knowledge gaps remain. Most of the research to date has focused on in vitro and animal models, leaving critical questions about the optimal dosage, timing, and long-term safety of melatonin supplementation in humans unanswered [9,84,92]. Future preclinical and clinical trials should systematically evaluate dosage regimens, treatment durations, and population-specific factors such as age, sex, exposure level, and comorbidities. Additionally, exploring synergistic combinations with other antioxidants or therapeutic agents may enhance melatonin’s efficacy. Long-term studies will be essential to establish its safety profile and therapeutic value in mitigating BPA-related health risks [9,84,92,95,96].

## 5. Potential Applications of Melatonin for BPA-Induced Toxicity

### 5.1. Melatonin Dosage and Timing in Animal Models

An optimal melatonin dosage is critical for achieving therapeutic efficacy against BPA-induced toxicity [93]. Although precise human-equivalent doses await clinical determination, animal studies offer valuable guidance. In murine models, melatonin at 10–20 mg/kg markedly attenuated oxidative stress and testicular damage following BPA exposure [93]. Specifically, Qi et al. demonstrated that a 20 mg/kg dose significantly increased testicular and epididymal indices, enhanced seminiferous tubule development, improved sperm viability and density, and upregulated both MT1 and luteinizing hormone receptor expression in Leydig cells [93]. Meanwhile, a 10 mg/kg regimen effectively reduced mitochondrial lipid peroxidation and restored the activity of key antioxidant enzymes, elevating GSH, SOD, and CAT levels while lowering MDA and H_2_O_2_ in BPA-exposed mice, thereby stimulating testosterone synthesis and reducing testicular cell apoptosis [93].

Dosage requirements may vary by target tissue and exposure severity, and lower doses have also been shown to have protective effects under certain conditions [92]. Consequently, future studies must identify optimal melatonin regimens for humans, considering age, sex, metabolic profile, and BPA exposure intensity [92]. The timing of administration is equally important: melatonin can serve as a prophylactic agent when administered before or during BPA exposure, preserving mitochondrial function, reducing MDA accumulation, and boosting GSH activity in pretreated animals [92]. Such preexposure dosing harnesses melatonin’s role as a first-line antioxidant to prevent the initiation of oxidative damage [92].

Post-exposure treatment with melatonin likewise confers therapeutic benefits. Melatonin has been shown to restore sperm motility and quality, and protect adrenal and prostate function in male rats when administered after BPA challenge. These effects are attributed to the capacity of melatonin to repair oxidative damage to DNA, lipids, and proteins [90,97,98]. In vitro work further substantiates that melatonin pretreatment maintains mitochondrial integrity under BPA stress [92,94]. Thus, the selection of pre- versus post-exposure administration may depend on the specific health outcome and individual exposure profile [92].

Melatonin’s efficacy in preventing and treating BPA toxicity hinges on its modulation of molecular pathways: it enhances the expression of antioxidant enzymes, scavenges free radicals, and regulates inflammatory responses [92,93]. Its lipophilic nature facilitates rapid cellular and mitochondrial uptake, promoting the detoxification of ROS and RNS while activating endogenous defenses [92]. These actions are both dose- and time-dependent: higher doses may be necessary to induce robust enzyme expression, whereas lower doses suffice for immediate radical scavenging [92]. Melatonin also stimulates glutathione S-transferase (GST) activity, thereby enhancing BPA biotransformation and reducing bioaccumulation [99].

To optimize clinical application, it is essential to delineate melatonin’s dose–response relationships and timing strategies. Pretreatment appears to be particularly effective at forestalling oxidative stress, whereas postexposure administration may better support cellular recovery [92]. Although compelling animal and in vitro evidence underpins melatonin’s protective role, well-designed human trials are imperative. Future research should evaluate dose–response effects across diverse populations with varying exposure levels, explore synergistic combinations with other antioxidants, and conduct long-term clinical studies assessing endpoints such as body weight, testosterone levels, sperm quality, and testicular weight [84,92].

In summary, melatonin exhibits substantial promise as both a preventive and therapeutic agent against BPA toxicity. Its dosage and timing are pivotal in modulating key molecular pathways and mitigating BPA-induced damage. Although further clinical validation is needed, current preclinical findings support the translation of melatonin into strategies for reducing the health risks associated with BPA exposure [92].

### 5.2. Safety Considerations in Human

Melatonin is generally recognized as safe, with a low incidence of serious adverse effects, as reported in animal and human studies [100]. Animal models demonstrated exceptionally low acute melatonin toxicity, underscoring its favorable safety profile [101]. However, most of the available data are derived from short-term studies, and the long-term safety of melatonin supplementation remains unclear.

Short-term melatonin use is considered safe [102]. The most commonly reported adverse effects include drowsiness, headache, dizziness, and nausea, which are typically mild and transient and resolve upon discontinuation [102,103]. Higher doses may induce more pronounced effects, such as confusion, disorientation, and vivid dreams or nightmares [102]. Therefore, adherence to recommended doses and the ongoing monitoring of adverse reactions are essential [102].

Certain populations, including children, pregnant women, and the elderly, require special consideration for melatonin use [84,104]. Melatonin is increasingly being used to manage sleep disorders in pediatric populations, particularly in children with neurodevelopmental conditions [102]. Although short-term studies suggest general safety in children, concerns persist regarding the long-term effects on hormonal development, especially the timing of puberty [102,105]. Evidence suggests that prolonged melatonin use may delay puberty onset [105,106]. Consequently, pediatric use should be guided by healthcare professionals with careful developmental monitoring [102,105]. The use of melatonin in children under two years of age is typically discouraged because of insufficient safety data [102,105]. Additionally, the growing availability of melatonin in flavored gummy formulations has raised concerns regarding accidental pediatric overdoses [102,103,105].

Melatonin use during pregnancy is another area of uncertainty [104]. Although preliminary studies indicate potential benefits, including reduced risks of preeclampsia and preterm birth, data on long-term fetal safety are limited [107,108]. Melatonin is naturally synthesized in placental tissue and plays a role in the implantation and maintenance of pregnancy [104]. Nonetheless, melatonin is generally not recommended during pregnancy unless explicitly prescribed by a healthcare provider owing to the lack of comprehensive safety data [104,105,106].

Melatonin supplementation in older adults warrants caution due to its potential effects on balance and postural control. Postural stability notably decreased following melatonin administration, which increased the risk of falls in older individuals [109,110,111]. Age-related changes in melatonin metabolism may alter its efficacy and duration of action, requiring individualized dose adjustments [110,112].

Melatonin may interact with various medications, influencing efficacy or increasing the risk of adverse effects [113]. For example, medications such as tricyclic antidepressants, fluvoxamine, cimetidine, and oral contraceptives inhibit melatonin metabolism, thereby elevating serum melatonin levels [113]. Conversely, drugs such as carbamazepine, omeprazole, and beta-blockers may reduce circulating melatonin concentrations [109,113].

Certain contraindications and precautions should be considered. Individuals with autoimmune diseases should use melatonin cautiously, as melatonin may stimulate immune activity [110,114]. Although the findings are inconsistent, some reports suggest that melatonin increases seizure frequency in individuals with epilepsy, whereas others report either no effect on or a reduction in seizures [110,115,116,117]. Individuals with cardiovascular conditions should be monitored closely when using melatonin due to its potential effects on blood pressure [110,118]. High doses of melatonin have occasionally been associated with exacerbated psychological symptoms in patients with depressive disorders [110,119].

Although melatonin shows promise in reducing BPA toxicity, the safety profile of melatonin must be thoroughly evaluated, particularly in sensitive groups such as children, pregnant women, and the elderly [9,84,104]. Professional consultation is strongly advised before initiating supplementation in these groups. Responsible melatonin use, guided by clinical evidence and dosing recommendations, is essential to maximize benefits and minimize potential risks [9,102].

Future studies should address several critical gaps to further clarify the safety of melatonin in the context of BPA toxicity [92,120]. Further studies are necessary to evaluate the effects of prolonged melatonin supplementation on hormonal balance, reproductive health, and neurological outcomes. Clinical trials should examine the safety and efficacy of melatonin administration across diverse demographic groups, considering variables such as age, sex, and preexisting health conditions. In addition, further investigation into the potential interactions between melatonin and medications commonly used in individuals exposed to BPA is needed. Studies should also explore the effect of melatonin on biomarkers of BPA-induced toxicity, such as oxidative stress and endocrine hormone levels.

## 6. Comprehensive Research Framework

### 6.1. Summary of Findings

This review explored the current evidence on the protective effects of melatonin against BPA toxicity. BPA is a widespread endocrine-disrupting chemical associated with a range of adverse health effects, primarily owing to its ability to disrupt hormone signaling, induce oxidative stress, and trigger inflammatory responses. Melatonin is a neurohormone with potent antioxidant and anti-inflammatory properties that has shown promising protective effects against BPA-induced toxicity in numerous in vitro and animal studies.

These results suggest that melatonin attenuates BPA-induced reproductive toxicity, reduces oxidative damage, protects cellular integrity, and modulates inflammation. Table 3 and Figure 3 summarize representative findings on the antioxidant effects of melatonin across various organ systems, including the kidneys, testes, reproductive system, and brain, as reported in peer-reviewed scientific articles published between 2020 and 2025.

The emerging evidence also highlights additional mechanisms through which melatonin may counteract BPA toxicity. These mechanisms include its capacity to chelate transition metal ions, limiting free radical production, and its ability to preserve cell membrane integrity. Moreover, melatonin directly modulates mitochondrial function, supports energy production, and reduces mitochondrial ROS generation. The influence of melatonin on gene expression, particularly on genes involved in antioxidant defense and cellular protection, further underscores the multifaceted role of melatonin in mitigating BPA-induced damage.

### 6.2. Implications for Public Health and Policy

The preclinical findings indicate that melatonin is a promising intervention for mitigating the health risks associated with widespread BPA exposure, particularly reproductive dysfunction and endocrine disruption. Melatonin supplementation could play a large role in public health strategies, including preventive recommendations for individuals at heightened risk of BPA exposure, should clinical studies validate these protective effects in human populations. These outcomes may also inform regulatory policies by supporting stricter limitations on BPA use in consumer products and highlighting the potential of melatonin as a countermeasure. Ongoing research is crucial to enhance our understanding of the health effects of BPA and address the wider challenges associated with environmental exposure to endocrine-disrupting chemicals.

### 6.3. Future Research Directions

Recent research trends have revealed new ways in which melatonin exerts protective effects, extending beyond its established antioxidant and anti-inflammatory actions. This research aims to enhance our understanding of the therapeutic potential of melatonin in mitigating the effects of BPA toxicity.

#### 6.3.1. Modulation of Specific Reproductive Signaling Pathways

Recent studies indicate that melatonin can directly counteract BPA’s interference with female reproductive function by restoring specific signaling pathways in granulosa cells (GCs). BPA has been shown to downregulate the expression of the follicle-stimulating hormone receptor (FSHR) and inhibit the FSH-induced expression of Connexin 43 (Cx43), which are crucial for normal folliculogenesis and oocyte maturation. Melatonin treatment effectively reverses this downregulation of FSHR and restores Cx43 expression, thereby mitigating BPA-induced toxicity in GCs. This suggests a more targeted endocrine modulation by melatonin beyond general cellular protection.

#### 6.3.2. Inhibition of ER Stress Pathways

Melatonin has been found to significantly alleviate BPA-induced testicular apoptosis and ER stress. Melatonin directly inhibits ER stress-related pathways by decreasing the levels of key proteins such as p-PERK, p-IRE1, and ATF6α, which are activated during ER stress. This inhibition leads to the suppression of testicular cell apoptosis and the enhancement of testosterone synthesis.

BPA is known to induce cellular stress, including ER stress, which can lead to apoptosis and dysfunction in various tissues, including the testes. Melatonin’s ability to specifically target and inhibit these ER stress pathways offers a precise protective mechanism.

#### 6.3.3. Regulation of Autophagy via Transcription Factor EB (TFEB) and p38 MAPK Pathway

Melatonin has been shown to mitigate neuronal damage by upregulating autophagy through the promotion of nuclear translocation of TFEB. TFEB is a master regulator of lysosomal biogenesis and autophagy. Additionally, melatonin improves BPA-induced cell apoptosis, oxidative stress, and autophagy impairment by inhibiting the p38 MAPK signaling pathway.

BPA can impair autophagy and induce apoptosis and oxidative stress. Melatonin’s direct influence on TFEB and the p38 MAPK pathway provides a novel and specific way to restore cellular clearance mechanisms and reduce cell death.

#### 6.3.4. Inhibition of Ferroptosis Through KEAP1/NRF2/PTGS2 Axis and Ferritinophagy

Melatonin has been identified as an inhibitor of ferroptosis, a distinct form of regulated cell death characterized by iron-dependent lipid peroxidation. It attenuates ferroptosis by activating the KEAP1/NRF2/PTGS2 signaling pathway, leading to the upregulation of antioxidant enzymes like glutathione peroxidase 4 (GPx4) and cystine/glutamate antiporter (xCT), and the downregulation of pro-ferroptotic proteins like KEAP1 and PTGS2. Furthermore, melatonin alleviates cadmium-induced ferroptosis (a relevant toxicological model) by regulating ferritinophagy (the selective degradation of ferritin, an iron-storage protein) and iron metabolism, specifically by inhibiting NCOA4-mediated ferritinophagy in spermatogonia.

While the direct link to BPA-induced ferroptosis is still being explored, BPA is known to induce oxidative stress and can lead to various forms of cell death. BPA has also been reported to induce ferroptosis in renal injury models. Melatonin’s specific anti-ferroptotic mechanisms offer a new therapeutic target against BPA-induced cellular damage.

#### 6.3.5. Regulation of Mitochondrial Dynamics and Quality Control via SIRT1/PGC-1α Pathway

Beyond general mitochondrial protection, melatonin actively regulates mitochondrial quality control, encompassing mitochondrial biogenesis, dynamics (fission and fusion), and mitophagy (autophagy of mitochondria). Specifically, melatonin promotes the expression of PGC-1α and activates the SIRT1 signaling pathway. This leads to increased expression of NRF2 and its downstream antioxidant genes, mitochondrial respiratory chain complex-related genes, mitochondrial biogenesis genes, and mitochondrial fusion genes, while significantly reducing mitochondrial fission genes.

BPA is known to impair mitochondrial function and induce mitochondrial dysfunction. Melatonin’s ability to restore mitochondrial homeostasis by modulating their dynamics and biogenesis through specific pathways like SIRT1/PGC-1α represents a sophisticated protective mechanism.

#### 6.3.6. New Methods in Future Research

Single-Cell RNA Sequencing: This advanced technique is being employed to gain detailed insights into the regulatory mechanisms governing cellular processes, such as the synchronized dormancy and activation between granulosa cells and oocytes. Applying this method to BPA-exposed cells treated with melatonin could reveal precise gene expression changes at a single-cell resolution, offering a deeper understanding of melatonin’s protective pathways.

Molecular Docking Simulations: Used to predict interactions between melatonin and specific proteins or receptors, such as KEAP1, in the context of ferroptosis. This computational method can help to identify potential direct molecular targets of melatonin and guide further experimental validation.

Nanomedicine for Targeted Delivery: Given melatonin’s hydrophobic nature, low water solubility, and short half-life, nanomedicine approaches are being explored to improve its bioavailability and enable targeted delivery, particularly to the central nervous system. Developing melatonin-loaded nanomaterials could enhance their therapeutic efficacy against BPA-induced neurotoxicity and other systemic effects.

Multi-Omics Approaches: While not explicitly linked to BPA and melatonin in the provided snippets, the use of multi-omics (e.g., metagenomics, metabolomics, and transcriptomics) is highlighted as a cutting-edge approach to comprehensively assess how interventions modulate the gut microbiome and its metabolites. Applying this to BPA toxicity could reveal complex interactions between BPA, the microbiome, and melatonin’s protective effects.

It is hypothesized that these novel mechanisms and advanced methods will serve as important pioneers in understanding and harnessing the protective capabilities of melatonin against the pervasive threat of BPA toxicity. It is anticipated that future research will facilitate the integration of these approaches, thereby contributing to the development of more effective therapeutic strategies.

## 7. Conclusions

This review investigated the protective effects of melatonin against BPA toxicity, drawing from recent in vitro, animal, and limited preclinical studies. BPA is identified as a pervasive endocrine-disrupting chemical widely used in consumer products, leading to widespread human exposure, primarily through dietary supplements, and the use of dietary supplements. The disease has been linked to adverse health outcomes such as reproductive dysfunction, developmental abnormalities, metabolic disorders, neurodevelopmental impacts, and increased susceptibility to hormone-sensitive cancers. BPA exerts its toxicity by mimicking or antagonizing natural hormones through interactions with estrogen, androgen, and thyroid hormone receptors, as well as influencing G-protein-coupled estrogen receptors. Additionally, it triggers oxidative stress, compromises mitochondrial function, causes epigenetic alterations, and induces various forms of cell death, including apoptosis, necroptosis, ferroptosis, and autophagy. The immune system is particularly sensitive to BPA exposure, with a lowering of the tolerable daily intake due to concerns about immune toxicity (Figure 1).

Melatonin, a neuroendocrine hormone produced mainly by the pineal gland, exhibits potent antioxidant and anti-inflammatory properties in the brain. It directly scavenges free radicals and enhances the activity of endogenous antioxidant enzymes like SOD, CAT, and GPx, and it can be used to treat inflammatory diseases. Melatonin also modulates inflammatory responses by inhibiting pro-inflammatory cytokines and enzymes, and promoting anti-inflammatory macrophage polarization. Beyond these factors, melatonin influences epigenetic regulation, mitochondrial dynamics, and various cellular signaling pathways, contributing to its broad protective effects (Figure 2 and Figure 3).

In conclusion, the existing evidence strongly suggests that melatonin holds significant promise as a therapeutic candidate for mitigating the adverse effects of BPA exposure. Its protective actions are mediated through multiple biological pathways, including antioxidant defense, inflammation reduction, mitochondrial preservation, and support for reproductive function. However, despite these promising in vitro, animal, and preclinical findings, the current preclinical evidence for the effectiveness of melatonin against BPA toxicity remains limited. Comprehensive human clinical trials should confirm these findings with further research, establish optimal dosing and timing, and evaluate the long-term safety and efficacy of melatonin as an intervention to counteract BPA toxicity.

## Figures and Tables

**Figure 1 ijms-26-07526-f001:**
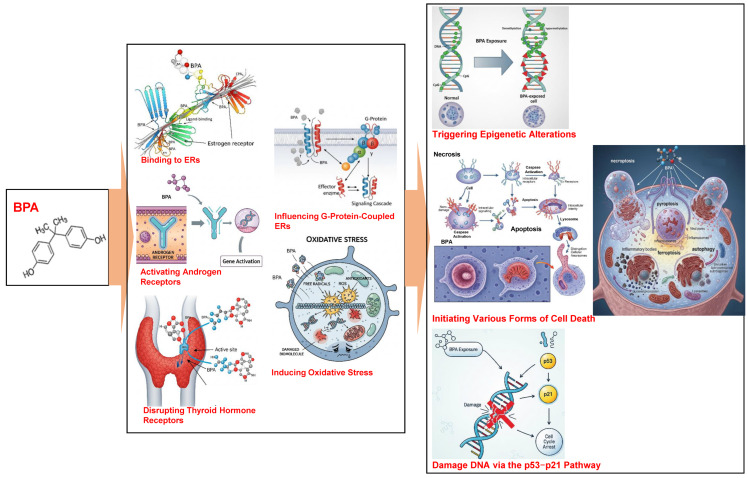
The mechanisms of BPA toxicity. ERs, estrogen receptors.

**Figure 2 ijms-26-07526-f002:**
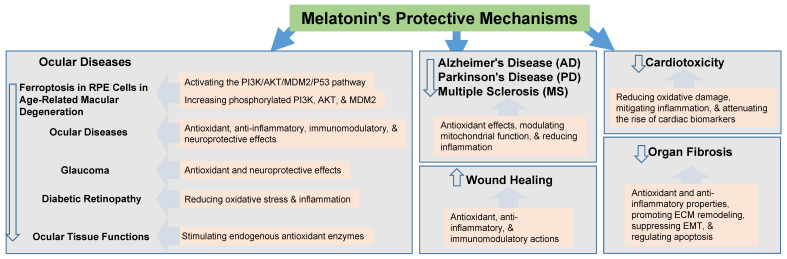
The protection mechanism of melatonin.

**Figure 3 ijms-26-07526-f003:**
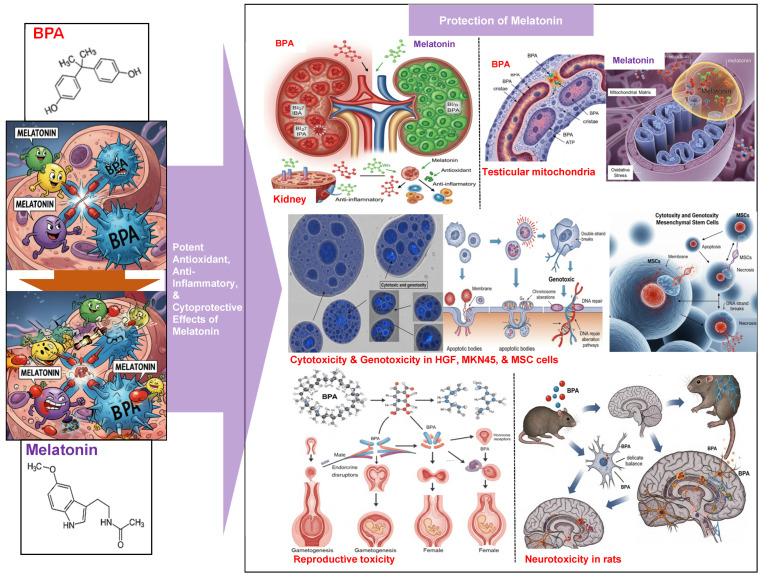
The protective effect of melatonin against the toxicity of BPA.

**Table 1 ijms-26-07526-t001:** Summary of the mechanisms behind the development of different types of cancer following BPA exposure (2020–2025).

Cancer Type	Specificity	Proposed Mechanism	BPA Concentration	Reference
Breast Cancer	General *	Endocrine disruption; promoting adipogenesis, lipogenesis, and adipokine secretion; creating a pro-inflammatory, nutrient-rich environment	Not specified	[36]
	ERα-positive cells	Synergizes with genistein to induce estrogenic responses, potentially epigenetically reprogramming breast cells	50 nM	[37]
	General	Induces differentiation of adipose cells into cancer-associated adipocyte-like cells, promoting epithelial–mesenchymal transition (EMT) via the CXCL12/AKT pathway	10 nM	[38]
Prostate Cancer	General	Promotes biochemical recurrence and death by disrupting mitochondrial energy homeostasis, potentially via ESR1-PFKFB4 axis	In vitro and in vivo: 10 nM	[39]
Ovarian Cancer	General	Promotes migration and invasion by activating the Wnt/β-catenin/SPP1 axis, leading to osteopontin secretion and transformation of fibroblasts into cancer-associated fibroblasts	In vitro: 10, 100 nM	[40]
	General	Alters epithelial diversity; induces apoptosis and necrosis; disrupts antioxidant, apoptotic, and inflammatory gene expression	Low dose 1 mg/kg body weight (BW), high dose 5 mg/kg BW	[41]
	General	Induces oxidative stress, increasing production of ovarian cancer stem cells	Low dose 1 mg/kg BW, high dose 5 mg/kg BW	[42]
Endometrial Cancer	General	Low-dose BPA alters estrous cycle and uterine pathology in rats, with a gene signature predictive of survival in human patients with endometrial cancer	25 and 250 μg/kg BW/day	[43]
Colorectal Cancer	General	Enhances de novo ceramide synthesis, exacerbating tumor progression and EMT	In vitro: 0.01, 0.1, 1 µM; in vivo: low dose 0.62–1.14 μg/g, high dose 6.65–20.56 μg/g	[44]
	Obese rats	May worsen progression through the PI3K–AKT pathway and increase IL-1β levels	25 mg/kg	[45]
	General	Upregulates *GOLPH3*, promoting proliferation and migration	1 µM	[46]
	Colon epithelial cells	Increases cellular invasiveness and anchorage-independent cell growth, potentially through phosphorylation of various protein kinases	Low dose 0.0043 nM	[47]
Thyroid Cancer	Papillary thyroid carcinoma	Enhances proliferation and tumorigenesis through ROS generation and activation of NOX4 signaling pathways	0.1 and 0.5 µM	[48]
Liver Cancer	Hepatocellular carcinoma (in mice)	Alters gene expression in the liver, predicting hepatocellular carcinoma	50 mg/kg	[49]
	Liver cells	Induces chemosensitivity and is associated with increased PPARγ expression in digestive system cancers	~33.70 µg/mL	[50]

* The term ‘General’ is used to indicate a comprehensive study or identification of a cancer type that is not limited to a specific subtype or focused on a specific cell line. Specifically, it means that the findings are applicable across cancer types and not limited to a specific variant or cell type.

**Table 2 ijms-26-07526-t002:** Summary of the protective effects of melatonin against diseases and other conditions (2020–2025).

	Specificity of Protective Effect	Proposed Mechanism	Melatonin Concentration	Reference
Ocular Diseases	Protects against ferroptosis in RPE cells in age-related macular degeneration	PI3K/AKT/MDM2/p53 pathway: increases phosphorylated PI3K, AKT, and MDM2, inhibiting p53 and restoring SLC7A11 expression	In vitro: 10–200 μM; in vivo: 10–40 mg/kg	[73]
	General protective and therapeutic potential for ocular diseases	Antioxidant, anti-inflammatory, immunomodulatory, neuroprotective, regulation of intraocular pressure, and VEGF secretion	Varies depending on disease and study	[74]
	IOP-lowering and neuroprotection in glaucoma	Stimulation of melatonin receptors in the ciliary body; antioxidant and neuroprotective effects	Topical formulations: 0.1–17.2 mM	[75]
	Antioxidant and neuroprotective effects in diabetic retinopathy	Reduction in oxidative stress and inflammation	Varies depending on study	[74]
	Antioxidant protection and regulates ocular tissue functions	Direct scavenging of ROS, stimulation of antioxidant enzymes, interaction with melatonin receptors	0.07–86 mM	[75]
Neurodegenerative Diseases	Protective activity in Alzheimer’s disease, Parkinson’s disease, and multiple sclerosis	Antioxidant, modulates mitochondrial function and inflammation, synergistic effects with other neuroprotective agents	Optimal dosage has not yet been established in clinical settings	[76]
Protection Against Electromagnetic Waves	Protects various organs (brain, skin, eyes, testis, kidney) against cell phone-induced electromagnetic waves	Strengthens cellular antioxidant system, mitigates oxidative stress and cell death	Animal studies: 2–100 mg/kg	[77]
Wound Healing	Increases general wound healing (cuts, burns, ulcers)	Antioxidant, anti-inflammatory, and immunomodulatory actions, regulates vascular reactivity and angiogenesis	Physiological (pM) to pharmacological (μM) levels	[78]
Skin Aging	Protection against age-related skin deterioration	Combats oxidative stress, shields from UV damage, curbs melanin production, and influences collagen synthesis and mitochondrial activity	Ex vivo: 100–200 µM	[79]
	Differential antiaging effects on epidermis and dermis	Downregulates mTORC1 activity and MMP-1, increases VEGF-A and fibrillin-1 expressions	Ex vivo: 100–200 µM	[80]
Cardiotoxicity	Protects against 5-fluorouracil-induced cardiotoxicity	Reduces oxidative damage, mitigates inflammation, attenuates the increase in cardiac biomarker levels	Rats: 2.5, 5, and 10 mg/kg/day	[81]
Organ Fibrosis	Inhibition of fibrosis in liver, lung, heart, kidney, and skin	Antioxidant, anti-inflammatory effects, remodels extracellular matrix, suppresses EMT, regulates apoptosis	Oral administration: up to 500 mg/day	[7]
Other Protective Effects	Mitigation of lead-induced oxidative stress in tobacco BY-2 suspension cells	Protects against lipid profile modification and membrane integrity	Not specified	[82]
	Increase in sleep quality in adults with various diseases	Not specified	Varies depending on study	[83]

**Table 3 ijms-26-07526-t003:** Summary of antioxidant effects of melatonin against BPA toxicity (2020–2025).

Specificity	Proposed Mechanism	Melatonin Concentration	BPA Concentration	Functions	References
Renal protection	Diminishing oxidative stress, maintaining redox equilibrium within mitochondria, sustaining mitochondrial function and architecture	In vivo: 10 mg/kg; in vitro preincubation: 0.1, 0.5, 1, 10, 100 µM; in vitro after BPA exposure: 0.5 µM	In vivo: 50, 100, 150 mg/kg; in vitro: 1–1000 µM; in vitro with melatonin: 125 µM	Damaging effects of BPA on the kidney and the protection by melatonin: emerging evidence from in vivo and in vitro studies	[94]
Protection of testicular mitochondria	Antioxidant properties, direct free radical scavenging activity, lowering mitochondrial lipid peroxidation, restoring mitochondrial enzyme activity, and ameliorating decreased enzymatic and nonenzymatic antioxidants	In vivo: 1–10 mg/kg BW	In vivo: 10 mg/kg BW; environmentally relevant BPA doses: 25 μg/kg	Melatonin ameliorates BPA-induced biochemical toxicity in testicular mitochondria of mice	[95,121]
Protection against cytotoxicity and genotoxicity in HGF, MKN45, and MSC cell lines	Antioxidant and free radical scavenging properties, reducing ROS and MDA levels, increasing GSH content, and preventing DNA damage	In vitro: 12, 23, 46, and 93 µg/mL	In vitro: 0.5, 5, 50, and 100 µg/mL; control for melatonin protection: 50 µg/mL	The primary route of human exposure to BPA is oral intake, which is associated with genotoxicity, oxidative stress, endocrine disruption, mutagenicity, and carcinogenicity in in vitro and in vivo models	[9]
Protection against reproductive toxicity	Increases in sperm concentration, sperm viability, and testosterone serum levels, potentially through antioxidant properties and interaction with hormone receptors, antiapoptotic effects, and hormonal modulation	Varies across studies: 10–20 mg/kg BW	Varies across included studies: 25–250 μg/kg BW	Melatonin and vitamins as protectors against the reproductive toxicity of bisphenols	[84,93]
Neuro-protection in rats	Reduction in oxidative stress (diminished MDA levels), modulation of ERK/NF-kB signaling pathway, attenuation of histopathological alterations in the hippocampus, improvement in behavioral changes	In vivo, oral administration: 20 mg/kg BW (MEL I), 40 mg/kg BW (MEL II)	In vivo, intraperitoneal administration: 1 mg/kg BW	Neurotoxicity of BPA and impact of melatonin administration on oxidative stress, ERK/NF-kB signaling pathway, and behavior in rats	[70,71]

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
