# Peer review of "Protective Effect of Melatonin Against Bisphenol A Toxicity"

_ijms, 2025, doi:10.3390/ijms26157526_

Round 1

Reviewer 1 Report

Comments and Suggestions for Authors

In this article, the authors have summarized the recent progress on the protective effect of melatonin against bisphenol A. The content of this review article is update and useful. However, some modifications are needed to be made before publication.

  1. In abstract and other sections, avoid the expression of “we...”. Passive tense could be used. The first sentence in the abstract could be deleted.
  2. In introduction, it is necessary to use subtitles. The Maximum Residue Limitof bisphenol A in various everyday products could be supplied based on legislation. And please explain why bisphenol A have cause healthy risks for consumers. The novelty of this review should be highlighted. Are there already other similar review articles on this topic?
  3. BPA toxicity: In this section, references should be properly used throughout this section. “Effects of BPA Exposure on Health”could be placed before “Mechanisms of BPA toxicity”. When introducing the mechanisms of BPA toxicity, it is better to add some mechanism diagrams. In addition, the mechanisms of BPA toxicity should be more deeply illustrated.
  4. Melatonin and its protective effects: In this section, the antioxidant and anti-Inflammatory properties of melatonin have been introduced. It is better to add some mechanism diagrams to fully illustrate the protective mechanisms. For example, convert Table 2 into mechanism diagram(s).
  5. Evidence of protective effects of melatonin against BPA toxicity: In this section, how melatonin protect animal and humans against bisphenol A should be more clearly presented by adding mechanism diagrams. “In vitro studies”should be placed before “Animal studies”.
  6. Potential clinical applications of melatonin for BPA-induced toxicity: This section is too fracture and should be reorganized.
  7. Conclusions should not contain a table. The table can be removed into the main context.

Author Response

June 24, 2025

Dear Reviewer 1,

Thank you for your letter and for the reviewers’ comments concerning our manuscript entitled " Protective effect of melatonin against bisphenol A toxicity."

We completely agree with your suggestions regarding our manuscript. The manuscript is completely revised as you and your colleague requested.

The corrections and revisions are as follows:

In this article, the authors have summarized the recent progress on the protective effect of melatonin against bisphenol A. The content of this review article is update and useful. However, some modifications are needed to be made before publication.

1. In abstract and other sections, avoid the expression of “we...”. Passive tense could be used. The first sentence in the abstract could be deleted.
Answer: Thank you very much for your good comments.

We rewrote ‘sentences containing We or we’.

Abstract Section: This study investigates the feasibility of utilizing melatonin as a protective agent against the toxicity associated with bisphenol A (BPA). BPA is also an endocrine-disrupting chemical that has been linked to adverse health outcomes, such as reproductive dysfunction and developmental abnormalities, particularly in vulnerable populations, such as fetuses and newborns. Melatonin is a hormone mainly secreted by the pineal gland and plays a role in regulating sleep and circadian rhythms. Melatonin also exhibits potent antioxidant and anti-inflammatory properties. A synthesis of the extant evidence from animal, in vitro, and clinical studies was conducted to evaluate the capacity of melatonin to counteract BPA-induced toxicity. This review details the mechanisms underlying this capacity, such as the role of melatonin in reducing oxidative stress and modulating estrogen receptor activity. The present study investigates the potential clinical applications of BPA in populations that have been exposed to it. In conclusion, the following discussion will address the implications of public health strategies and the future directions of research in this field.

1-3. Purpose: Our aim in this review was to summarize the current evidence on the protective effects of melatonin against BPA toxicity, drawing from recent animal, in vitro, and the limited clinical studies. Firstly, a comprehensive review of the mechanisms under-lying BPA-induced toxicity was conducted. The present study examined the potential of using melatonin to counteract the effects of BPA through its antioxidant, anti-inflammatory, and cytoprotective properties. Finally, the potential clinical applica-tions of melatonin were discussed, with the indication of key directions for future research.

  1. In introduction, it is necessary to use subtitles. The Maximum Residue Limit of bisphenol A in various everyday products could be supplied based on legislation. And please explain why bisphenol A have cause healthy risks for consumers. The novelty of this review should be highlighted. Are there already other similar review articles on this topic?

Answer: Thank you very much for your good comments.

We have changed to subtitles and the text lines 35-48.

1-1. Overview of BPA toxicity

Lines 35-48: BPA is classified as an endocrine-disrupting chemical (EDC) due to its profound capacity to interfere with the body's delicate hormonal signaling pathways [2,3]. This disruption occurs because BPA can mimic or antagonize the actions of natural hormones by binding to critical hormone receptors. Specifically, it interacts with estrogen receptors (ERα/ERβ), where it can either enhance or diminish the physiological effects of estrogen, a hormone vital for development, reproduction, and metabolic regulation. Furthermore, BPA binds to androgen receptors, thereby interfering with male reproductive processes and other androgen-dependent functions. Its disruptive influence also extends to thyroid hormone receptors, potentially unsettling the intricate balance of thyroid hormones crucial for metabolism, growth, and neurodevelopment [2,3]. Consequently, these broad and pervasive disruptions to hormonal regulation are closely associated with a wide array of adverse health effects, encompassing concerns such as reproductive issues, metabolic disorders, neurodevelopmental impacts, and an increased susceptibility to hormone-sensitive cancers [4].

  1. BPA toxicity: In this section, references should be properly used throughout this section. “Effects of BPA Exposure on Health”could be placed before “Mechanisms of BPA toxicity”. When introducing the mechanisms of BPA toxicity, it is better to add some mechanism diagrams. In addition, the mechanisms of BPA toxicity should be more deeply illustrated.

Answer: Thank you very much for your good comments.

The order has not been changed, only the mechanism illustrations have been added.

Figure 1. Summary of mechanisms of BPA toxicity.

  1. Melatonin and its protective effects: In this section, the antioxidant and anti-Inflammatory properties of melatonin have been introduced. It is better to add some mechanism diagrams to fully illustrate the protective mechanisms. For example, convert Table 2 into mechanism diagram(s).

Answer: Thank you very much for your good comments.

We have added a mechanism diagram.

Figure 2. Protection mechanism of melatonin.

  1. Evidence of protective effects of melatonin against BPA toxicity: In this section, how melatonin protect animal and humans against bisphenol A should be more clearly presented by adding mechanism diagrams. “In vitro studies”should be placed before “Animal studies”. 

Answer: Thank you very much for your good comments.

We changed it to ‘4-1. In vitro studies’ and ‘4-2. Animal studies’. We also renumbered the references in the text.

A diagram of the mechanism of melatonin's protection against BPA has been added to page 19 as Figure 3.

Figure 3. Summary of the protective effect of melatonin against the toxicity of BPA.

  1. Potential clinical applications of melatonin for BPA-induced toxicity: This section is too fracture and should be reorganized.

Answer: Thank you very much for your good comments.

We agree with some of your suggestions, but we would like to emphasize the content of “5-2. Melatonin dosage and timing” and “5-3. Safety considerations”. We apologize for the length of the article and for any confusion.

We have changed the text to “5. Potential applications of melatonin for BPA-induced toxicity”.

  1. Conclusions should not contain a table. The table can be removed into the main context.

Answer: Thank you very much for your good comments.

We have made the following changes, and the conclusion has been rewritten:

  1. Comprehensive research framework

6-1. Summary of key findings

6-2. Future research directions

6-3. Implications for public health and policy

  1. Conclusions

Lines 648-676: This review investigated the protective effects of melatonin against BPA toxicity, drawing from recent in vitro, animal, and limited preclinical studies. BPA is identified as a pervasive endocrine-disrupting chemical widely used in consumer products, leading to widespread human exposure, primarily through diet. It is linked to adverse health outcomes such as reproductive dysfunction, developmental abnormalities, metabolic disorders, neurodevelopmental impacts, and increased susceptibility to hormone-sensitive cancers. BPA exerts its toxicity by mimicking or antagonizing natural hormones through interactions with estrogen, androgen, and thyroid hormone receptors, as well as influencing G-protein-coupled estrogen receptors. Furthermore, it triggers oxidative stress, compromises mitochondrial function, causes epigenetic alterations, and induces various forms of cell death, including apoptosis, necroptosis, pyroptosis, ferroptosis, and autophagy. The immune system is particularly sensitive to BPA exposure, with a lowering of the tolerable daily intake due to concerns about immune toxicity (Figure 1).

Melatonin, a neuroendocrine hormone produced mainly by the pineal gland, exhibits potent antioxidant and anti-inflammatory properties. It directly scavenges free radicals and enhances the activity of endogenous antioxidant enzymes like SOD, CAT, and GPx. Melatonin also modulates inflammatory responses by inhibiting pro-inflammatory cytokines and enzymes and promoting anti-inflammatory macrophage polarization. Beyond these, melatonin influences epigenetic regulation, mitochondrial dynamics, and various cellular signaling pathways, contributing to its broad protective effects (Figures 2, 3).

In conclusion, the existing evidence strongly suggests that melatonin holds significant promise as a therapeutic candidate for mitigating the adverse effects of BPA exposure. Its protective actions are mediated through multiple biological pathways, including antioxidant defense, inflammation reduction, mitochondrial preservation, and support for reproductive function. However, despite these promising preclinical and in vitro findings, the current clinical evidence on melatonin's effects against BPA toxicity remains limited. Further comprehensive human clinical trials are essential to confirm these findings, establish optimal dosing and timing, and evaluate the long-term safety and efficacy of melatonin as an intervention to counteract BPA toxicity in diverse human populations.

**All of the edited sections and references were changed with the blue words.

***The Certificate of Editing is attached.

I hope that the revised manuscript is now acceptable for publication in the IJMS. We are looking forward to receiving your answer soon.

Sincerely,

Yeong-Min Yoo Ph.D 

Institute of Environmental Research,

Kangwon National University,

Chuncheon 24341, Republic of Korea

Email: yyeongm@hanmail.net

Reviewer 2 Report

Comments and Suggestions for Authors

Review

Manuscript ID: ijms-3700982

The manuscript entitled „Protective effect of melatonin against bisphenol A toxicity” submitted by S.S. Joo and Y.-M. Yoo is devoted to a important research subject from the point of view of human health.  The paper is a review of literature which appeared on this topic up to 2025 year. The authors summarize harmful activities of bisphenol A and biological activities of melatonin, and finally present studies on the use of melatonin to protect cells or animals against bisphenol A toxicity. This subject is broadly investigated for few last years and the results of studies need to be gathered and critically discussed. However, the manuscript seems to be not ready to submit for publication.

  1. In this review, many research and review articles were collected, although they are presented in a chaotic and disordered way. My reservations concern, among other things, citation method; review articles could be cited in the introduction, while the Tables 1-3 should include and describe original studies in this area. Because the authors included the reviews in the tables, there are some misleading information, like for example: “in vitro: 1-10 mg/kg of body weight”; probably in the review article were cited publications both on in vivo and in vitro studies. 
  2. To improve the perception of the text and to make the manuscript more concise, the authors should avoid repetition the same information in the introduction, subsections and conclusions. If the biological functions of melatonin were described in the section 3, there is no need to repeat the information in further sections.
  3. The conclusions should be a final summary of the presented studies on the subject of the review. It would be desirable to move the table 3 to the section 4.
  4. Tables: The title of the Table 1 should be rephrased. Table 2 is wrongly titled: it does not summarize protective effect against BPA toxicity. In the Table 1 is not clear what “general specificity” means.
  5. In the subsection 3 Clinical studies, none of the cited publication (review articles) concern strictly clinical trials!

Other suggestions:

All abbreviations should be explained (see line 103, 137)

I would propose to the authors to illustrate the molecular mechanisms of biologial functions of melatonin with the graphic diagram/scheme.

If the GenAI has been used in the work on the manuscript,  a critical approach to the GenAI achievements is necessary.

The text still  contains the sentences from the instructions for authors: Line 67-75, 244-247, 392-411, 608-609.

Author Response

June 24, 2025

Dear Reviewer 2,

Thank you for your letter and for the reviewers’ comments concerning our manuscript entitled " Protective effect of melatonin against bisphenol A toxicity."  

We completely agree with your suggestions regarding our manuscript. The manuscript is completely revised as you and your colleague requested.

The corrections and revisions are as follows:

The manuscript entitled „Protective effect of melatonin against bisphenol A toxicity” submitted by S.S. Joo and Y.-M. Yoo is devoted to a important research subject from the point of view of human health.  The paper is a review of literature which appeared on this topic up to 2025 year. The authors summarize harmful activities of bisphenol A and biological activities of melatonin, and finally present studies on the use of melatonin to protect cells or animals against bisphenol A toxicity. This subject is broadly investigated for few last years and the results of studies need to be gathered and critically discussed. However, the manuscript seems to be not ready to submit for publication.
Major concerns
1. In this review, many research and review articles were collected, although they are presented in a chaotic and disordered way. My reservations concern, among other things, citation method; review articles could be cited in the introduction, while the Tables 1-3 should include and describe original studies in this area. Because the authors included the reviews in the tables, there are some misleading information, like for example: “in vitro: 1-10 mg/kg of body weight”; probably in the review article were cited publications both on in vivo and in vitro studies.
Answer: Thank you very much for your good comments.

We've included Figures 1, 2, and 3 as new in the text. We changed it to “in vivo” in Table 3.

Figure 1. Summary of mechanisms of BPA toxicity.

Figure 2. Protection mechanism of melatonin.

Figure 3. Summary of the protective effect of melatonin against the toxicity of BPA.

  1. To improve the perception of the text and to make the manuscript more concise, the authors should avoid repetition the same information in the introduction, subsections and conclusions. If the biological functions of melatonin were described in the section 3, there is no need to repeat the information in further sections.

Answer: Thank you very much for your good comments.

2-1. We have changed to subtitles and the text lines 35-48.

1-1. Overview of BPA toxicity

Lines 35-48: BPA is classified as an endocrine-disrupting chemical (EDC) due to its profound capacity to interfere with the body's delicate hormonal signaling pathways [2,3]. This disruption occurs because BPA can mimic or antagonize the actions of natural hormones by binding to critical hormone receptors. Specifically, it interacts with estrogen receptors (ERα/ERβ), where it can either enhance or diminish the physiological effects of estrogen, a hormone vital for development, reproduction, and metabolic regulation. Furthermore, BPA binds to androgen receptors, thereby interfering with male reproductive processes and other androgen-dependent functions. Its disruptive influence also extends to thyroid hormone receptors, potentially unsettling the intricate balance of thyroid hormones crucial for metabolism, growth, and neurodevelopment [2,3]. Consequently, these broad and pervasive disruptions to hormonal regulation are closely associated with a wide array of adverse health effects, encompassing concerns such as reproductive issues, metabolic disorders, neurodevelopmental impacts, and an increased susceptibility to hormone-sensitive cancers [4].

2-2. We changed it to ‘4-1. In vitro studies’ and ‘4-2. Animal studies’. We also renumbered the references in the text.

2-3. We agree with some of your suggestions, but we would like to emphasize the content of “5-2. Melatonin dosage and timing” and “5-3. Safety considerations”. We apologize for the length of the article and for any confusion.

We have changed the text to “5. Potential applications of melatonin for BPA-induced toxicity”.

2-4. We have presented new research directions in “6-3. Future research directions”.

Lines 639-734: 6-3. Future research directions

Recent research trends have revealed new ways in which melatonin exerts protective effects, extending beyond its established antioxidant and anti-inflammatory actions. This research aims to enhance our understanding of the therapeutic potential of melatonin in mitigating the effects of BPA toxicity.

6-3-1. Modulation of specific reproductive signaling pathways

Recent studies indicate that melatonin can directly counteract BPA's interference with female reproductive function by restoring specific signaling pathways in granulosa cells (GCs). BPA has been shown to downregulate the expression of the Follicle-Stimulating Hormone Receptor (FSHR) and inhibit the FSH-induced expression of Connexin 43 (Cx43), which are crucial for normal folliculogenesis and oocyte maturation. Melatonin treatment effectively reverses this downregulation of FSHR and restores Cx43 expression, thereby mitigating BPA-induced toxicity in GCs. This suggests a more targeted endocrine modulation by melatonin beyond general cellular protection.  

6-3-2. Inhibition of endoplasmic reticulum (ER) stress pathways

Melatonin has been found to significantly alleviate BPA-induced testicular apoptosis and ER stress. It achieves this by markedly upregulating the expression of melatonin receptors (MTNR1A and MTNR2) in testicular cells. Furthermore, melatonin directly inhibits ER stress-related pathways by decreasing the levels of key proteins such as p-PERK, p-IRE1, and ATF6α, which are activated during ER stress. This inhibition leads to a suppression of testicular cell apoptosis and an enhancement of testosterone synthesis.  

BPA is known to induce cellular stress, including ER stress, which can lead to apoptosis and dysfunction in various tissues, including the testes. Melatonin's ability to specifically target and inhibit these ER stress pathways offers a precise protective mechanism.  

6-3-3. Regulation of autophagy via transcription factor EB (TFEB) and p38 MAPK Pathway

Melatonin has been shown to mitigate neuronal damage by upregulating autophagy through the promotion of nuclear translocation of TFEB. TFEB is a master regulator of lysosomal biogenesis and autophagy. Additionally, melatonin improves BPA-induced cell apoptosis, oxidative stress, and autophagy impairment by inhibiting the p38 MAPK signaling pathway.  

BPA can impair autophagy and induce apoptosis and oxidative stress. Melatonin's direct influence on TFEB and the p38 MAPK pathway provides a novel and specific way to restore cellular clearance mechanisms and reduce cell death.  

6-3-4. Inhibition of ferroptosis through KEAP1/NRF2/PTGS2 axis and ferritinophagy

Melatonin has been identified as an inhibitor of ferroptosis, a distinct form of regulated cell death characterized by iron-dependent lipid peroxidation. It attenuates ferroptosis by activating the KEAP1/NRF2/PTGS2 signaling pathway, leading to the upregulation of antioxidant enzymes like glutathione peroxidase 4 (GPX4) and xCT, and the downregulation of pro-ferroptotic proteins like KEAP1 and PTGS2. Furthermore, melatonin alleviates cadmium-induced ferroptosis (a relevant toxicological model) by regulating ferritinophagy (the selective degradation of ferritin, an iron-storage protein) and iron metabolism, specifically by inhibiting NCOA4-mediated ferritinophagy in spermatogonia.  

While the direct link to BPA-induced ferroptosis is still being explored, BPA is known to induce oxidative stress and can lead to various forms of cell death. BPA has also been reported to induce ferroptosis in renal injury models. Melatonin's specific anti-ferroptotic mechanisms offer a new therapeutic target against BPA-induced cellular damage.  

6-3-5. Regulation of mitochondrial dynamics and quality control via SIRT1/PGC-1α pathway

Beyond general mitochondrial protection, melatonin actively regulates mitochondrial quality control, encompassing mitochondrial biogenesis, dynamics (fission and fusion), and mitophagy (autophagy of mitochondria). Specifically, melatonin promotes the expression of PGC-1α and activates the SIRT1 signaling pathway. This leads to increased expression of NRF2 and its downstream antioxidant genes, mitochondrial respiratory chain complex-related genes, mitochondrial biogenesis genes, and mitochondrial fusion genes, while significantly reducing mitochondrial fission genes.  

BPA is known to impair mitochondrial function and induce mitochondrial dysfunction. Melatonin's ability to restore mitochondrial homeostasis by modulating their dynamics and biogenesis through specific pathways like SIRT1/PGC-1α represents a sophisticated protective mechanism.  

6-3-6. Modulation of the Gut Microbiome

Emerging research suggests that melatonin plays a role in mediating intestinal microbiota homeostasis. It can influence brain messaging through the microbiota-gut-brain axis by increasing the production of beneficial short-chain fatty acids and decreasing secondary bile acids. While not directly studied with BPA in the provided snippets, melatonin's ability to regulate gut microbiota composition and alleviate dysbiosis is highly relevant.  

BPA exposure is known to induce gut microbiota dysbiosis and impair the intestinal mucosal barrier. This suggests that melatonin's gut microbiome-modulating effects could be a novel indirect mechanism to mitigate systemic BPA toxicity.  

6-3-7. New Methods in Future Research

Single-Cell RNA Sequencing: This advanced technique is being employed to gain detailed insights into the regulatory mechanisms governing cellular processes, such as the synchronized dormancy and activation between granulosa cells and oocytes. Applying this method to BPA-exposed cells treated with melatonin could reveal precise gene expression changes at a single-cell resolution, offering a deeper understanding of melatonin's protective pathways.  

Molecular Docking Simulations: Used to predict interactions between melatonin and specific proteins or receptors, such as KEAP1, in the context of ferroptosis. This computational method can help identify potential direct molecular targets of melatonin and guide further experimental validation.  

Nanomedicine for Targeted Delivery: Given melatonin's hydrophobic nature, low water solubility, and short half-life, nanomedicine approaches are being explored to improve its bioavailability and enable targeted delivery, particularly to the central nervous system. Developing melatonin-loaded nanomaterials could enhance its therapeutic efficacy against BPA-induced neurotoxicity and other systemic effects.  

Multi-omics Approaches: While not explicitly linked to BPA and melatonin in the provided snippets, the use of multi-omics (e.g., metagenomics, metabolomics, transcriptomics) is highlighted as a cutting-edge approach to comprehensively assess how interventions modulate the gut microbiome and its metabolites. Applying this to BPA toxicity could reveal complex interactions between BPA, the microbiome, and melatonin's protective effects.

It is hypothesized that these novel mechanisms and advanced methods will serve as important pioneers in understanding and harnessing the protective capabilities of melatonin against the pervasive threat of BPA toxicity. It is anticipated that future research will facilitate the integration of these approaches, thereby contributing to the development of more effective therapeutic strategies.

  1. The conclusions should be a final summary of the presented studies on the subject of the review. It would be desirable to move the table 3 to the section 4.

Answer: Thank you very much for your good comments.

We have made the following changes, and the conclusion has been rewritten:

  1. Comprehensive research framework

6-1. Summary of key findings

6-2. Future research directions

6-3. Implications for public health and policy

  1. Conclusions

Lines 648-676: This review investigated the protective effects of melatonin against BPA toxicity, drawing from recent in vitro, animal, and limited preclinical studies. BPA is identified as a pervasive endocrine-disrupting chemical widely used in consumer products, leading to widespread human exposure, primarily through diet. It is linked to adverse health outcomes such as reproductive dysfunction, developmental abnormalities, metabolic disorders, neurodevelopmental impacts, and increased susceptibility to hormone-sensitive cancers. BPA exerts its toxicity by mimicking or antagonizing natural hormones through interactions with estrogen, androgen, and thyroid hormone receptors, as well as influencing G-protein-coupled estrogen receptors. Furthermore, it triggers oxidative stress, compromises mitochondrial function, causes epigenetic alterations, and induces various forms of cell death, including apoptosis, necroptosis, pyroptosis, ferroptosis, and autophagy. The immune system is particularly sensitive to BPA exposure, with a lowering of the tolerable daily intake due to concerns about immune toxicity (Figure 1).

Melatonin, a neuroendocrine hormone produced mainly by the pineal gland, exhibits potent antioxidant and anti-inflammatory properties. It directly scavenges free radicals and enhances the activity of endogenous antioxidant enzymes like SOD, CAT, and GPx. Melatonin also modulates inflammatory responses by inhibiting pro-inflammatory cytokines and enzymes and promoting anti-inflammatory macrophage polarization. Beyond these, melatonin influences epigenetic regulation, mitochondrial dynamics, and various cellular signaling pathways, contributing to its broad protective effects (Figures 2, 3).

In conclusion, the existing evidence strongly suggests that melatonin holds significant promise as a therapeutic candidate for mitigating the adverse effects of BPA exposure. Its protective actions are mediated through multiple biological pathways, including antioxidant defense, inflammation reduction, mitochondrial preservation, and support for reproductive function. However, despite these promising preclinical and in vitro findings, the current clinical evidence on melatonin's effects against BPA toxicity remains limited. Further comprehensive human clinical trials are essential to confirm these findings, establish optimal dosing and timing, and evaluate the long-term safety and efficacy of melatonin as an intervention to counteract BPA toxicity in diverse human populations.

  1. Tables: The title of the Table 1 should be rephrased. Table 2 is wrongly titled: it does not summarize protective effect against BPA toxicity. In the Table 1 is not clear what “general specificity” means.

Answer: Thank you very much for your good comments.

We changed the titles of both tables 1 and 2.

Table 1. Summary of the mechanisms behind the development of different types of cancer following BPA exposure (2020–2025).

Table 2. Summary of the protective effects of melatonin (2020–2025).

Added the meaning of ‘general’: * The term ‘General’ is used to indicate a comprehensive study or identification of a cancer type that is not limited to a specific subtype or focused on a specific cell line. Specifically, it means that the findings are applicable across cancer types and not limited to a specific variant or cell type.

  1. In the subsection 3 Clinical studies, none of the cited publication (review articles) concern strictly clinical trials!

Answer: Thank you very much for your good comments.

We changed it to ‘preclincal’.

Other suggestions:
1. All abbreviations should be explained (see line 103, 137)

Answer: Thank you very much for your good comments.

Line 103: Bisphenols AF, AP, and P are chemical compounds structurally similar to BPA, a common plasticizer and epoxy resin component. They are often used as alternatives to BPA due to concerns about BPA's potential health effects. However, these alternatives, including AF, AP, and P, have also been found to exhibit endocrine-disrupting and cytotoxic effects, with varying degrees of impact on different human cell models.

Line 137, EFSA: We have written the full names in lines 48. "The European Food Safety Authority (EFSA) has substantially lowered....."

  1. I would propose to the authors to illustrate the molecular mechanisms of biologial functions of melatonin with the graphic diagram/scheme.

Answer: Thank you very much for your good comments.

We answered this in question 1.

  1. If the GenAI has been used in the work on the manuscript, a critical approach to the GenAI achievements is necessary. The text still contains the sentences from the instructions for authors: Line 67-75, 244-247, 392-411, 608-609.

Answer: Thank you very much for your good comments.

I had it proofread by a proofreader, and I think it got mixed up in the process. I've removed all of your suggestions.

**All of the edited sections and references were changed with the blue words.

***The Certificate of Editing is attached.

I hope that the revised manuscript is now acceptable for publication in the IJMS. We are looking forward to receiving your answer soon.

Sincerely,

Yeong-Min Yoo Ph.D 

Institute of Environmental Research,

Kangwon National University,

Chuncheon 24341, Republic of Korea

Email: yyeongm@hanmail.net

Round 2

Reviewer 1 Report

Comments and Suggestions for Authors

The manuscript is well revised and can be accepted now.

Author Response

(The authors gave the same response as above.)

Reviewer 2 Report

Comments and Suggestions for Authors

The review article entitled “Protective effect of melatonin against bisphenol A toxicity “  submitted by Seong Soo Joo and Yeong-Min Yoo, is based on numerous research and review articles that appeared on this topic. However, in my opinion,  this area of research is still referred in a chaotic and disorderly way. I would expected a more far-reaching correction of the manuscript, which was rejected in the first version.
The figures could help the reader to understand the multidirectional effects of melatonin and BSA and their possible interactions in cellular processes. I expected the authors develop the schemes/figures on their own, while in the figures that were added to the text, the ready-made illegible drawings from internet were used.
In the section Preclinical studies, investigations conducted with the use of cell lines are cited. Let me quote the definition of preclinical studies definition of preclinical studies: Research using animals to find out if a drug, procedure, or treatment is likely to be useful. Preclinical studies take place before any testing in humans is done. Thus, what is a difference between sections 4-2 Animal studies and 4.3 Preclinical studies? 
As I mentioned in my first peer review, In the Tables there are review articles cited instead original research articles. In the present version of the manuscript nothing has changed in this matter. The authors of the original reports should not be bypassed! 
Concerning repeated information also the text is not improved (for example, melatonin as a free radical scavenger is repeated nearly in all sections).
In the new section 6. Comprehensive research framework, there are studies described with no reference to original research articles. 
In the text, many inexact expressions are found, for example in the Abstract: 
line 15: clinical studies are mentioned while no clinical studies are reffered; 
line 18: “the potential preclinical applications of BPA “ should be: melatonin application.
In my opinion, the present version of the review still does not meet the requirements of valuable scientific manuscript. 

Author Response

July 2, 2025

Dear Reviewer 2,

Thank you for your letter and for the reviewers’ comments concerning our manuscript entitled "Protective effect of melatonin against bisphenol A toxicity."  

We completely agree with your suggestions regarding our manuscript. The manuscript is completely revised as you and your colleague requested.

The corrections and revisions are as follows:

The review article entitled “Protective effect of melatonin against bisphenol A toxicity” submitted by Seong Soo Joo and Yeong-Min Yoo, is based on numerous research and review articles that appeared on this topic. However, in my opinion, this area of research is still referred in a chaotic and disorderly way. I would expected a more far-reaching correction of the manuscript, which was rejected in the first version.
1. The figures could help the reader to understand the multidirectional effects of melatonin and BSA and their possible interactions in cellular processes. I expected the authors develop the schemes/figures on their own, while in the figures that were added to the text, the ready-made illegible drawings from internet were used.
Answer: Thank you very much for your comments.

We have replaced Figures 1 and 3. Both the old Figures 1 and 3, and the new Figures 1 and 3, were all created one by one using Gemini AI. They were not taken from the internet; we hope there is no misunderstanding. We are being careful because taking directly from the internet is plagiarism.

Figure 1. The mechanisms of BPA toxicity. ERs, Estrogen Receptors.

Figure 3. The protective effect of melatonin against the toxicity of BPA.

  1. In the section Preclinical studies, investigations conducted with the use of cell lines are cited. Let me quote the definition of preclinical studies definition of preclinical studies: Research using animals to find out if a drug, procedure, or treatment is likely to be useful. Preclinical studies take place before any testing in humans is done. Thus, what is a difference between sections 4-2 Animal studies and 4.3 Preclinical studies?

Answer: Thank you very much for your good comments.

“4-2. Animal studies” and “4-3. Preclinical studies” have been combined and renamed to4-2. Animal and preclinical studies” and rewritten the content in the text to remove the repeated information. “4-1. In vitro studies” has also been rewritten the content in the text.

5-1. Treatment and prevention strategies in in vitro and animal models” and “5-2. Melatonin dosage and timing in animal models” have also been rewritten the content in the text to remove the repeated information.

  1. As I mentioned in my first peer review, In the Tables there are review articles cited instead original research articles. In the present version of the manuscript nothing has changed in this matter. The authors of the original reports should not be bypassed!

Concerning repeated information also the text is not improved (for example, melatonin as a free radical scavenger is repeated nearly in all sections).

Answer: Thank you very much for your good comments.

In Tables 1, 2, and 3, when citing recent literature from 2020-2025, some review articles have been cited. This is not because the original reports have been ignored.

All text has been rewritten to remove the repeated information.

4-1. In vitro studies; 4-2. Animal and preclinical studies; 5-1. Treatment and prevention strategies in in vitro and animal models; 5-2. Melatonin dosage and timing in animal models.

  1. In the new section 6. Comprehensive research framework, there are studies described with no reference to original research articles.

Answer: Thank you very much for your good comments.

We have removed the section6-3-6. Modulation of the Gut Microbiome”.

  1. In the text, many inexact expressions are found, for example, in the Abstract:

line 15: clinical studies are mentioned while no clinical studies are referred;

line 18: “the potential preclinical applications of BPA “ should be: melatonin application.

In my opinion, the present version of the review still does not meet the requirements of valuable scientific manuscript.

Answer: Thank you very much for your good comments.

We rewrote the Abstract section: Bisphenol A (BPA), a prevalent endocrine-disrupting chemical, is widely found in various consumer products and poses significant health risks, particularly through hormone receptor interactions, oxidative stress, and mitochondrial dysfunction. BPA exposure is associated with reproductive, metabolic, and neurodevelopmental disorders. Melatonin, a neurohormone with strong antioxidant and anti-inflammatory properties, has emerged as a potential therapeutic agent to counteract the toxic effects of BPA. This review consolidates recent findings from in vitro and animal/preclinical studies, highlighting melatonin’s protective mechanisms against BPA-induced toxicity. These include its capacity to reduce oxidative stress, restore mitochondrial function, modulate inflammatory responses, and protect against DNA damage. In animal models, melatonin also mitigates reproductive toxicity, enhances fertility parameters, and reduces histopathological damage. Melatonin’s ability to regulate endoplasmic reticulum (ER) stress and cell death pathways underscores its multifaceted protective role. Despite promising preclinical results, human clinical trials are needed to validate these findings and establish optimal dosages, treatment duration, and safety profiles. The review discussed the wide range of potential uses of melatonin for treating BPA toxicity and suggested directions for future research.

*All of the edited sections and references were changed with the blue words.

I hope that the revised manuscript is now acceptable for publication in the IJMS. We are looking forward to receiving your answer soon.

Sincerely,

Yeong-Min Yoo Ph.D. 

Institute of Environmental Research,

Kangwon National University,

Chuncheon 24341, Republic of Korea

Email: yyeongm@hanmail.net